# Prevalence and Association of *Campylobacter* spp., *Salmonella* spp., and *Blastocystis* sp. in Poultry

**DOI:** 10.3390/microorganisms11081983

**Published:** 2023-08-01

**Authors:** Muriel Guyard-Nicodème, Nagham Anis, Doaa Naguib, Eric Viscogliosi, Marianne Chemaly

**Affiliations:** 1Unit for Hygiene and Quality of Poultry and Pork Products, Laboratory of Ploufragan-Plouzané-Niort, ANSES, F-22440 Ploufragan, France; naghamanis@gmail.com (N.A.); marianne.chemaly@anses.fr (M.C.); 2Department of Hygiene and Zoonoses, Faculty of Veterinary Medicine, Mansoura University, Mansoura 35516, Egypt; doaanaguib246@yahoo.com; 3CNRS, Inserm, CHU Lille, Institut Pasteur de Lille, U1019—UMR 9017—CIIL—Centre d’Infection et d’Immunité de Lille, University of Lille, F-59000 Lille, France; eric.viscogliosi@pasteur-lille.fr

**Keywords:** *Campylobacter*, *Salmonella*, *Blastocystis*, prevalence, interaction, poultry

## Abstract

Poultry and poultry meat are considered the most important sources of human campylobacteriosis and salmonellosis. However, data about the occurrence of *Campylobacter* and *Salmonella* concomitantly with intestinal protozoa such as *Blastocystis* sp. in poultry remain very scarce. Therefore, this study aimed to investigate the presence and possible interactions between these three microorganisms in fecal samples from 214 chickens collected either on farms or from live bird markets in Egypt. The results obtained showed that *Campylobacter* spp., *Salmonella* spp., and *Blastocystis* sp. were present in 91.6% (196/214), 44.4% (95/214), and 18.2% (39/214) of tested samples, respectively, highlighting an active circulation of these microorganisms. Moreover, a significant positive correlation was reported between the occurrence of *Campylobacter* spp. and *Blastocystis* sp. together with a significant negative correlation between *Blastocystis* sp. and *Salmonella* spp. This study confirms the association reported previously between *Blastocystis* sp. and *Campylobacter* spp. while disclosing an association between *Blastocystis* sp. and *Salmonella* spp.; it also highlights the need to improve studies on the interactions between bacteria and eukaryotes in the gut microbiota of poultry.

## 1. Introduction

*Campylobacter* is the leading pathogen behind foodborne disease in humans worldwide. According to the 2021 EFSA report, campylobacteriosis has been the most frequently reported foodborne disease in Europe since 2005, with 127,840 cases in 2021 [1]. This disease can lead to several symptoms such as fever, diarrhea, abdominal cramps, and vomiting that generally resolves within two to three weeks. In some cases, the illness can evolve toward more severe pathologies such as Guillain–Barré syndrome and reactive arthritis or, in rare cases, lead to the patient’s death [2,3]. Poultry is a major reservoir of *Campylobacter*, which plays the main role in transmitting this pathogen to humans [4,5]. Undercooked poultry meats such as broiler and turkey, along with cross-contamination during handling, are considered the main sources of human infection [6]. *Campylobacter* contamination in meat products is reflected by its high prevalence on farms and in slaughterhouses. The baseline survey conducted in Europe in 2008 showed a prevalence of *Campylobacter* spp. of 71.2% in live broilers [7]. In France, the prevalence of *Campylobacter* in ceca was of 77.2% [8]. In North Africa and especially Egypt, very few data are currently available on the burden of *Campylobacter* spp. in poultry populations. The few small-scale studies conducted in this country showed low overall prevalence rates depending on the nature of samples and the study design. *C. jejuni* was found in 16% of the cloacal swabs collected from one poultry farm [9] and in 18.12% [10] or 16.83% from incorporated broiler farms [11]. A higher prevalence of *Campylobacter* was reported in the manure storage area and broiler litter, with 66.7% and 53.3%, respectively [12].

In addition to *Campylobacter* spp., other enteric microorganisms including *Salmonella* spp. [13] and the protozoan *Blastocystis* sp. [14] can frequently colonize poultry. *Salmonella* spp. is an important foodborne pathogen found worldwide that causes salmonellosis [15]. *Blastocystis* sp. is the most frequently reported unicellular eukaryote colonizing both humans and many other animal hosts worldwide, and infection with this parasite in humans could be associated with various digestive disorders and urticaria [16]. Zoonotic potential of *Blastocystis* sp. from poultry has been demonstrated [17]. This anaerobic intestinal protozoan mainly colonizes the host’s large intestine [14,18], and the fecal–oral route is considered to be the main route of transmission to humans via direct contact with human or animals, or via contaminated water or food [14,18].

According to the 2007 EFSA report, the prevalence of *Salmonella* on broiler farms was of 23.7% in the EU, with a wide variation ranging from 0 to 68% between countries [19]. In France, epidemiological studies conducted in fecal samples of commercial poultry showed a prevalence of *Salmonella* spp. of 8.6% in broilers [20], 17.9% in laying hens [21], and 15.6% in fattening turkeys [22]. In Egypt, the prevalence of *Salmonella* in poultry at the farm level was reported in several studies, where it ranges from 2.5% in layers to 11.3% in broilers [23,24,25]. Few studies reported the prevalence of *Blastocystis* in poultry at the farm level, where it reached, respectively, 24 and 69.8% in two surveys conducted in Egypt [26,27].

Some previous studies have shown that the presence of *Campylobacter* in broilers could be associated with the presence of other microorganisms [28,29,30] and that it survived better under aerobic conditions in co-culture with other microorganisms such as *Pseudomonas* spp., *Acanthamoeba castellanii*, *Tetrahymena pyriformis*, *Staphylococcus aureus*, *Escherichia coli*, and *Salmonella* [31,32,33,34,35,36]. To date, no association between the occurrence of *Campylobacter* and *Salmonella* has been demonstrated on poultry farms, but few studies have been performed so far to confirm this hypothesis. Indeed, the simultaneous search for these two pathogens in new studies has recently been recommended in a recent EFSA opinion [37]. In addition, *Campylobacter* spp. has already shown a significant correlation with *Blastocystis* sp., suggesting that the presence of *Campylobacter* spp. may be promoted by the presence of *Blastocystis* sp. and, similarly, that the absence of one is associated with the absence of the other [38].

The aim of the present work was, thus, to assess the presence of *Campylobacter* spp., *Salmonella* spp., and *Blastocystis* sp. in poultry samples collected from farms and live bird markets in Egypt and to study the possible associations between these three pathogens concomitantly present in the samples.

## 2. Materials and Methods

### 2.1. Sample Collection

A total of 214 specimens were collected from poultry in four different Egyptian governorates (Al Dakahlia, Damietta, Kafr El Sheikh, and Gharbia) as part of a recent epidemiological study investigating the prevalence of *Blastocystis* sp. in various animal groups [39]. This study was approved by the Research Ethics Committee of the Faculty of Veterinary Medicine, Mansoura University, under code number R/99. Fecal samples from broilers, layers, and breeders were collected on the farm, and cecal samples were collected from broilers in live bird markets. Animals were aged from 14 to 539 days at the time of sampling as described in Table 1.

Briefly, five random droppings from various areas on the farms were pooled to compose a single specimen. For the live bird market specimens, five ceca from five randomly selected chickens belonging to the same batch were gathered, and cecal contents were recovered in a sterile plastic cup under completely aseptic conditions and pooled to form one specimen. Each sample was preserved in 2.5% potassium dichromate (half fecal material/half dichromate) and kept at 4 °C to prevent DNA deterioration prior to DNA extraction at Institut Pasteur of Lille, France.

### 2.2. DNA Extraction and qPCR

Fecal samples were washed three times in distilled water following centrifugation at 3000× *g* for 10 min, to eliminate the potassium dichromate. DNA was extracted from 500 µL of washed fecal samples using a commercial kit (NucleoSpin 96 Soil, Macherey-Nagel GmbH & Co KG, Düren, Germany) according to the manufacturer’s protocol. All specimens were examined for the presence of *Blastocystis* sp. with quantitative real-time PCR (qPCR) using 2 μL of extracted DNA as previously described [17]. All *Blastocystis* sp.-positive specimens were subtyped using sequence analysis of the purified qPCR products (Genoscreen, Lille, France).

A duplex qPCR was also performed for the simultaneous detection of *Campylobacter* spp. and *Salmonella* spp. in the same samples as described by Anis et al. [40]. This duplex PCR presented an amplification efficiency of 99.7 ± 1.5% and 97.7 ± 2.4% for *Campylobacter* and *Salmonella*, respectively, as previously determined [40]. The duplex qPCR reaction was performed in a final volume of 20 µL of reaction mix consisting of 10 µL PerfeCTa qPCR ToughMix (Quantabio, Beverly, MA, USA), 900 nM of each primer for *Campylobacter* spp., 100 nM of each primer for *Salmonella* spp., 125 nM of each probe with a cycle of amplification as follows: initial denaturation at 95 °C, 10 min followed by 40 cycles of 15 s at 95 °C and 1 min at 60 °C. The PCR assay was performed in duplicate with 2 µL of each extracted DNA sample and included positive (genomic DNAs for each target) and negative (non-template control) controls. Each reaction was considered positive for *Campylobacter* and *Salmonella* when the C_t_ (cycle threshold) was lower than 36.5 and 37, respectively.

### 2.3. Statistical Analysis

The prevalence of *Campylobacter* spp., *Blastocystis* sp., and *Salmonella* spp. was assessed, and the association between microorganisms was evaluated using the Fisher’s exact test or the chi-square test. The general significance level was set at a *p*-value (*p*) of below 0.05.

## 3. Results

### 3.1. Prevalence of Campylobacter spp., Salmonella spp., and Blastocystis sp. in Poultry Samples

Of the 214 samples, 196 tested positive for *Campylobacter* spp., 95 for *Salmonella* spp., and 39 for *Blastocystis* sp., which corresponded to an overall prevalence of 91.6%, 44.4% and 18.2%, respectively (Figure 1).

Table 2 presents the distribution of positive samples for the three microorganisms according to the origin of the samples.

When focusing on the farm level (Table 2), the results demonstrated that the prevalence for *Campylobacter* was not significantly different (*p* > 0.05) between broilers (89.8%; 44/49) and either layers (90.9%; 20/22) or breeders (87.5%; 14/16). In the same way, no significant difference in the prevalence of *Salmonella* (*p* > 0.05) was observed in broilers (44.9%; 22/49) compared with layers (50.0%; 11/22) or breeders (56.2%; 9/16). No significant difference was reported for *Blastocystis* sp., but the number of positive samples for this microorganism was limited (14 at the farm level). Additionally, no difference was observed in the distribution of the three microorganisms in broiler samples from the farm or the live bird market (*p* > 0.05), even considering that different samples (fecal from the farms or cecal from live bird markets) were analyzed. However, the total number of samples positive for *Blastocystis* sp. remained limited (34 broiler samples) (Table 2).

### 3.2. Assessment of the Association between Campylobacter spp., Salmonella spp., and Blastocystis sp. Infections in Poultry Samples

The correlation between the concomitant presence in the samples of *Campylobacter* spp., *Salmonella* spp., and *Blastocystis* sp. was investigated. The results, shown in Figure 2, indicated that the prevalence of co-contamination within the samples by *Campylobacter* spp. and *Salmonella* spp.; *Campylobacter* spp. and *Blastocystis* sp.; *Salmonella* spp. and *Blastocystis* sp.; and *Campylobacter* spp., *Salmonella* spp., and *Blastocystis* sp. was of 39.2%, 18.2%, 4.7%, and 4.7%, respectively.

No significant association was found between *Campylobacter* spp. and *Salmonella* spp. Indeed, among the 214 samples, 112 were positive only for *Campylobacter* spp., 11 were positive only for *Salmonella* spp., 84 were positive for both *Campylobacter* spp. and *Salmonella* spp., and seven samples were negative for both bacteria.

Table 3 presents the distribution of co-contaminated samples according to the origin of the samples and the poultry type.

At the farm level, co-contamination by *Campylobacter* spp. and *Salmonella* spp. was observed in 50% of the layers (11/22) and breeders (8/16) and in 38.8% (19/49) of the broilers. A similar rate of 36.2% (46/127) was observed for broilers from live bird markets (Table 3).

A significant positive association was found (*p* < 0.05) between *Campylobacter* spp. and *Blastocystis* sp. Indeed, all the samples contaminated by *Blastocystis* sp. were also contaminated by *Campylobacter* spp. At the farm level, broilers and layers had a similar co-contamination rate of 18.4% (9/49) and 18.2% (4/22), respectively, whereas the simultaneous contamination of breeders by *Campylobacter* spp. and *Blastocystis* sp. was observed only in 6.2% of the samples, representing only one out of six samples (Table 3). Broilers from live bird markets were co-contaminated at a rate of 19.7% (25/127), which is similar to broilers from poultry farms (Table 3).

A significant negative association was also highlighted (*p* < 0.05) between *Salmonella* spp. and *Blastocystis* sp., since the majority of samples contaminated by *Blastocystis* sp. were not positive for *Salmonella* spp. despite the high prevalence of *Salmonella* spp. in the samples. Indeed, only ten samples were co-infected with *Salmonella* spp. and *Blastocystis* sp. (Table 3). At the farm level, broilers and layers had a rate of co-infection by *Salmonella* spp. and *Blastocystis* sp. of 2.0% (1/49) and 9.1% (2/22), respectively, while no sample from breeders was positive for both these microorganisms. Broilers from live bird markets were co-contaminated at a rate of 5.5% (7/127) (Table 3).

## 4. Discussion

This study focused on poultry, which are considered a major source of human campylobacteriosis and salmonellosis worldwide. The aim of the survey was to assess the rate of contamination by *Campylobacter* spp. and *Salmonella* spp. of samples collected from farms and live bird markets in Egypt and investigate possible correlations between these two pathogens together and with the protozoan *Blastocystis* sp., colonizing poultry caecum [14,41,42].

*Campylobacter* spp. were detected in 91.6% of poultry samples, showing a high prevalence in all poultry types and confirming the potential public health risk represented by poultry. Studies investigating *Campylobacter* spp. prevalence in poultry in Egypt are scarce, and a comparison with the results of the present survey was not possible due to discrepancies in study design, sampling nature, and methodology. It is noteworthy that a prevalence of around 16% in cloacal swabs collected on poultry farms was recently documented [9,11]. However, a *Campylobacter* spp. prevalence of 81%, comparable with that of our survey, has been reported in samples collected from chicken ceca in another study conducted in Egypt [10].

In the present work, *Salmonella* spp. were detected in 44.4% of poultry samples, reflecting a high prevalence in all poultry types and highlighting this health risk for the human population. Furthermore, the reference method for the detection of *Salmonella* in poultry (EN ISO 6579-1) relies on a standard protocol of non-selective pre-enrichment followed by selective enrichment under aerobic conditions. However, not all the poultry samples analyzed herein underwent this pre-enrichment step, indicating that the observed prevalence is probably underestimated and that the samples were contaminated to a level sufficient to be detected with PCR without a pre-enrichment step. It was not possible to compare the results of the current study with other studies assessing *Salmonella* prevalence in poultry in Egypt because of discrepancies in study conditions, sampling matrices, and the method of analysis. However, a prevalence of around 10% was recently reported in broiler flocks sampled on poultry farms [24]. This low prevalence could be mainly due to the methodology, the low number of regions considered, the different ages of the investigated poultry, and the nature of sampling.

*Blastocystis* sp. was herein detected in 18.2% of poultry samples. This prevalence was, for instance, much lower than the infection rate of 69.8% reported in a cohort of farm chickens sampled in the Ismailia governorate [27]. Such a variation in prevalence can be explained by the different geographical locations and lifestyle of the animals. Indeed, the survey conducted in the Ismailia governorate included free-ranging chickens reared in rural areas where the risk of *Blastocystis* sp. infection is high through contaminated environmental sources.

In addition, pooled poultry samples were analyzed in the current study, which could result in a higher prevalence for the three microorganisms. Moreover, unlike microbiological methods, PCRs can detect dead microorganisms.

The assessment of the co-contamination of poultry samples by both *Campylobacter* spp. and *Salmonella* spp. showed high prevalence rates but no significant association between the occurrences of the two pathogens. This could be due to the fact that the majority of samples collected were positive for *Campylobacter* spp. and/or that the pre-enrichment step for *Salmonella* spp. detection was not performed in this work, leading to an underestimation of its prevalence. Indeed, in a recent study, a positive correlation between *Campylobacter* spp. and *Salmonella* spp. was demonstrated in which the survival of *Campylobacter* spp. in co-culture with *Salmonella* spp. under aerobic conditions was promoted [43]. Moreover, in a previous in vivo trial conducted in poultry, a correlation between the two pathogens was observed, with the colonization of *Salmonella* being enhanced by 1 log in the presence of *Campylobacter* [44]. These results also suggest that interactions are possible and probably common between these two microorganisms. Some previous studies have already highlighted a survivalist aspect of *C. jejuni* whereby it benefits from the presence of other microorganisms to survive under stressful aerobic conditions. The survival of *C. jejuni* under aerobic conditions was thus attributed either to a metabolic commensalism with the support of *Pseudomonas* spp. [34] or to the reduced levels of dissolved oxygen by *Acanthamoeba castellanii*, creating the microaerobic conditions for *C. jejuni* [35], or even to the metabolites produced by *Staphylococcus aureus* [36]. Therefore, a similar relationship may exist between *Campylobacter* spp. and *Salmonella* spp. that must be confirmed in further large-scale epidemiological surveys.

The co-contamination of poultry samples by both *Campylobacter* spp. and *Blastocystis* sp. in the current survey reflected a positive correlation between the two pathogens, demonstrating that the presence of *Blastocystis* sp. was significantly associated with the presence of *Campylobacter* spp. in all the poultry samples collected for this study. This positive correlation has already been described in a previous study performed in Lebanon which indicated that a large proportion (24.2%) of cecal samples were co-infected by *Blastocystis* sp. and *Campylobacter* spp., suggesting that the presence of *Campylobacter* spp. may be promoted by the presence of *Blastocystis* sp. and, similarly, that the absence of one is associated with the absence of the other [38]. No data are currently available regarding the impact of *Blastocystis* sp. on the bacterial intestinal microbiota of chickens, but based on our results and those previously published [38], it could be hypothesized that contamination by *Blastocystis* sp. could enhance the colonization of poultry by *Campylobacter* spp. and vice versa. Indeed, the *Blastocystis* sp. is increasingly recognized as an important component of the host gut microbiota and able to modulate the host’s immune response [45]. The correlations between *Blastocystis* sp. and other communities of intestinal microbiota have been investigated. Thus, the *Blastocystis* sp. plays a crucial role in the regulation of host–bacteria interactions in the gut and could facilitate colonization by other enteric pathogens such as *Campylobacter* spp. in infected birds. It could also explain the fact that all samples positive for *Blastocystis* sp. were positive for *Campylobacter* spp. In the same way, a positive effect of *Blastocystis* sp. on *E. coli* abundance was reported after co-infection in the fecal samples of mice [46]. Thus, future studies focused on poultry colonized by *Blastocystis* sp. are needed to shed light on the mechanisms involved in its association with *Campylobacter* spp., as, indeed, with other bacterial communities. In addition, several studies have demonstrated that the presence of *Campylobacter* spp. in poultry could be associated with the presence of other microorganisms, and that the diversity of poultry’s intestinal microbiome could be modified by the colonization of *Campylobacter* spp. [29,30]. For example, *C. jejuni* colonization was described to be associated with an increase in the relative abundance of *Bifidobacterium* [29], *Streptococcus*, and *Blautia* [47]. Moreover, an association of *C. jejuni* with the relative abundance of some taxa (*Escherichia*, *Alistipes*, *Enterococcus*, *Bacteroïdes*, *Shigella*, *Gallibacterium*, *Lactobacillus*, *Corynebacterium*, *Ruminococcaceae*, and *Enterobacter*) [47] and a positive correlation between *C. jejuni* and *Clostridium perfringens* in poultry ceca have also been highlighted [28,48]. All these data could explain the positive correlation between *Campylobacter* spp. and *Blastocystis* sp. demonstrated in the present study.

This study also revealed a significant negative correlation between the presence of *Salmonella* spp. and *Blastocystis* sp., wherein the majority of samples contaminated by *Blastocystis* sp. were not contaminated by *Salmonella* spp. This negative correlation between *Salmonella* spp. and *Blastocystis* sp. may be further studied, as it could be due to the underestimation of *Salmonella* spp. prevalence with the protocol used in this study. However, a similar negative correlation was observed in mice, with *Blastocystis* sp. having a negative effect on the abundance of *Bifidobacterium* and *Lactobacillus* [46], possibly explained by the lethal effect of oxidative stress together with other host factors induced by *Blastocystis* sp. [46]. This negative effect of *Blastocystis* sp. on *Bifidobacterium* was also supported by other studies [49] which demonstrated that the presence of *Blastocystis* was associated with a decrease in fecal protective bacteria such as *Faecalibacterium prausnitzii*, which is known for its anti-inflammatory properties. Consequently, the negative effect of *Blastocystis* sp. might be linked to the pathophysiology of irritable bowel syndrome with constipation (IBS-C) and intestinal flora imbalance [49].

## 5. Conclusions

In conclusion, this study showed a high prevalence of *Campylobacter* spp. and *Salmonella* spp. in Egyptian poultry fecal samples, suggesting that poultry represents a significant reservoir for human campylobacteriosis and salmonellosis. It highlighted a significant correlation between *Campylobacter* spp. and *Blastocystis* sp., demonstrating that the presence of *Blastocystis* sp. would be encouraged when *Campylobacter* spp. is present and vice versa. Furthermore, a significant negative correlation between *Salmonella* spp. and *Blastocystis* sp. was highlighted. These positive and negative correlations between *Blastocystis* sp. and *Campylobacter* spp. and *Salmonella* spp., respectively, need to be further studied to decipher the underlying mechanism of these associations. No correlation was found between the three pathogens together within the context of this investigation.

## Figures and Tables

**Figure 1 microorganisms-11-01983-f001:**
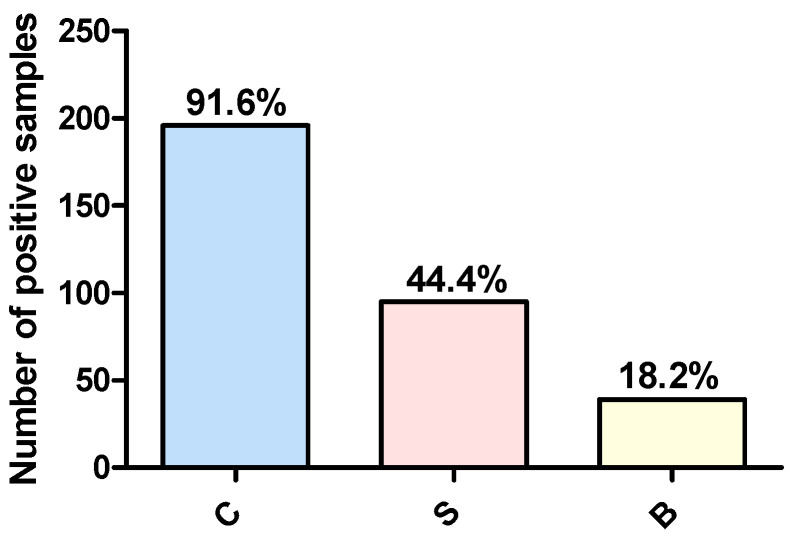
Prevalence of *Campylobacter* spp. (C), *Salmonella* spp. (S), and *Blastocystis* sp. (B) in the 214 poultry samples.

**Figure 2 microorganisms-11-01983-f002:**
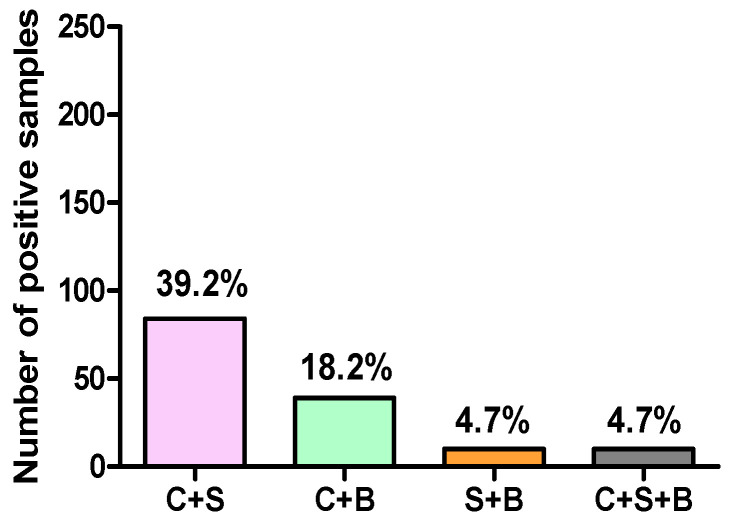
Prevalence of co-infection by *Campylobacter* spp. and *Salmonella* spp. (C + S), *Campylobacter* spp. and *Blastocystis* sp. (C + B), *Salmonella* spp. and *Blastocystis* sp. (S + B), and with all three microorganisms (C + S + B) in 214 poultry samples. Each bar presents the number of positive samples, and the corresponding proportion among all tested samples is presented in percentage above each bar.

**Table 1 microorganisms-11-01983-t001:** Characteristics and distribution of the poultry samples.

		Broilers	Layers	Breeders
Age (days)		14–116	132–510	194–539
Chicken breeds		Ross; Cobb; Hubbard; Sasso; Baladi	White; Brown	Cobb; Hubbard; White
Number of samples	Farm	49	22	16
	Live bird market	127	0	0
Total number of samples	214			

**Table 2 microorganisms-11-01983-t002:** Distribution of samples contaminated with *Campylobacter* spp. (C), *Salmonella* spp. (S), and *Blastocystis* sp. (B) according to the origin of the samples and poultry type.

			Number of Positive Samples		
Origin of the Samples	Poultry Type	Number of Samples	C	S	B
Farm	Broilers	49	44	22	9
	Layers	22	20	11	4
	Breeders	16	14	9	1
Live bird market	Broilers	127	118	53	25

**Table 3 microorganisms-11-01983-t003:** Distribution of samples co-contaminated with *Campylobacter* spp. and *Salmonella* spp. (C + S), *Campylobacter* spp. and *Blastocystis* sp. (C + B), *Salmonella* spp. and *Blastocystis* sp. (S + B), and with all three microorganisms (C + S + B) according to the origin of the samples and poultry type.

			Number of Positive Samples			
Origin of the Samples	Poultry Type	Number of Samples	C + S	C + B	S + B	C + S + B
Farm	Broilers	49	19	9	1	1
	Layers	22	11	4	2	2
	Breeders	16	8	1	0	0
Live bird market	Broilers	127	46	25	7	7

## Data Availability

Not applicable.

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
