# Peer review of "Prevalence and Association of Campylobacter spp., Salmonella spp., and Blastocystis sp. in Poultry"

_microorganisms, 2023, doi:10.3390/microorganisms11081983_

Round 1
Reviewer 1 Report
There are more recent zoonosis reports available. Replace the data or add the latest one you can find.
As it appears in the article, the tests were performed without a pre-enrichment step, which indicates that the samples were contaminated to a level that could be detected by PCR without a pre-enrichment step. This also means that the Salmonella prevalence is more likely higher than that, and therefore does not represent the true picture. This may also be a reason for the significant negative correlation between Salmonella spp. and Blastocystis sp. Part of the conclusion is already well known. Please estimate in some way what the result would have been if you had also done enrichment?
The chapter on materials and methods is incomplete. Please add the following information
- What positive and negative controls were used?
- Describtion of the determination of the effectiveness of the PCR.
Could it be possible that Campylobacter is present in samples containing Blastocystis sp. because it is intracellularly present, which was already hipothetised by some authors.
The results and conclusions of the article have been largely published in previous publications.
Reviewer 2 Report
Comments and Suggestions for Authors
This is a very significant study in an important subject, the prevalence and association of Campylobacter spp., Salmonella spp., and Blastocystis sp. in poultry.
The study is well design and the results support the conclusion. However, some issues should be addressed.
Page 4, Table 2: The Table should be before the results description. Immediately after “Table 2 presents the distribution of positive samples for the three microorganisms according to the origin of the samples.” The, after the Table the description make sense “When focusing on the farm level (Table 2), results demonstrated …”. Alternately, the authors should insert the % on the Table.
In my opinion if the Table is after the description it is bit strange because we already read the % and then we are having the absolute values?
Page 5, Figure 2: The scale and the bars should be changed. The graph is either in % or in absolute values. It makes no sense to have the axis in absolute values and the bars in %.
Page 5, Table 3: In this Table the authors should first present the Table and then analyse it (as in Table 2). The authors can add the % to the Table and in this case it can be in the same place. Alternately the Table should be presented and then described explaining the %.
This is a very elegant study, that after these changes should be publishable.
Reviewer 3 Report
|
The manuscript "Prevalence and association of Campylobacter spp., Salmonella spp., and Blastocystis sp. in poultry" by Muriel Guyard-Nicodème et al describes the prevalence of important food-borne human pathogens in poultry. 1. Can feces be used for association studies? I think either histology or culture may be a better option. 2. Blastocystis being an anaerobe may not have direct interactions with either Salmonella a facultative anaerobe or with Campylobacter which is microaerophilic (The authors have discussed this in the discussion part). Tissue biopsy and histology may be more useful to show interactions. 3. The authors may also include information on which part of the intestine these individual microorganisms colonize.
|
2 |
Reviewer 4 Report
The authors present a manuscript describing research that examined the prevalence and association of Campylobacter spp., Salmonella spp., and Blastocystis sp. in poultry. This work is important because Campylobacter is the leading bacterial cause of foodborne illness and Salmonella is in the top 3 or 4 causes of bacterial foodborne illness. Understanding the carriage rate of these pathogens in the poultry used for human food is important to understating the source and spread of foodborne pathogens and how we might better decrease their spread.
For the introduction section of the paper, the description of Campylobacter and Salmonella was sufficient. Overall the introduction was well written and well sources, however, in the introduction, more information needs to be included on Blastocystis and it’s impact on cases of foodborne illness before being accepted for publication.
For the methods section, it was well written. There was sufficient detail describing the experimental design and materials/methods used. The details included are sufficient that someone could repeat the authors work if needed.
Results section was well written. The data in figure 1 describe the prevalence of Campylobacter spp., Salmonella spp., and Blastocystis sp. in the 214 poultry samples. The results clearly show that most poultry harbor Campylobacter, and about half harbor Salmonella, and about 20% harbor Blastocystis.
Data in Table 2 describe the distribution of samples contaminated with Campylobacter spp., Salmonella spp., and Blastocystis sp. based on the origin of the samples and poultry type. This was helpful to understand what types of birds are harboring these pathogens.
Since there was significance carriage of these pathogens in the poultry population in this study it seemed logical to test if the poultry could be co-infection by more than one of the pathogens at a time. Data in figure 2 describe Prevalence of co-infection by Campylobacter spp. and Salmonella spp., Campylobacter spp. and Blastocystis sp., Salmonella spp. and Blastocystis sp., and with all three microorganisms in the 214 poultry samples. It was important to note that a significant number of birds were co-carriers of Campylobacter and Salmonella and Campylobacter and Blastrocystis.
Similar to Table 2, Table 3 data in Table 3 describe the distribution of samples co-contaminated with Campylobacter spp. and Salmonella spp., Campylobacter spp. and Blastocystis sp., Salmonella spp. and Blastocystis sp., and with all three pathogens according to the origin of the samples and poultry type. This was helpful to understand what types of birds are co-harboring these pathogens. The authors do a nice job describing the significance of this co-carriage in the discussion section.
Overall the discussion and conclusion sections were well-written. The authors did a good job describing the larger impact of poultry carrying these pathogens and which birds were most likely to be carriers. The discussion of the co-carriage of these pathogens and their influence on each other was in particular well done. I think the boarder impacts of this work was adequately described by the authors in this section.
